# Comparison of Pollen-Collecting Abilities between *Apis mellifera* L. and *Bombus terrestris* L. in the Oil Tree Peony Field

**Junyi Bao** [1], **Kaiyue Zhang** [1], **Xiangnan He** [1], **Zhanfeng Chen** [2], **Junying Wang** [2], **Chunling He** [3,*] **and Xiaogai Hou** [1,*]

1   College of Agriculture, Henan University of Science & Technology, Luoyang 471023, China; baojunyi311@126.com (J.B.); zhangkaiyue95@126.com (K.Z.); xiangnanhe19@163.com (X.H.)
2   Sanmenxia Muxian Biotechnology Co., Ltd., Sanmenxia 472000, China; chenzhanfeng115@163.com (Z.C.); mx17703987965@163.com (J.W.)
3   College of Horticulture and Plant Protection, Henan University of Science & Technology, Luoyang 471023, China
*   Correspondence: hechunling68@126.com (C.H.); hkdhxg@haust.edu.cn (X.H.); Tel.: +86-136-5387-3065 (X.H.)

**Abstract:** The lack of pollinators in the oil tree peony field was a serious problem for the seed yield. In the preliminary studies, we found that the seed rate of oil tree peonies was significantly improved by supplementing bees during the flowering stages in the pollination net room. In addition, the performance of *Apis mellifera* L. was better than *Bombus terrestris* L. To understand the interaction relationship between flowering characteristics and the managed bees, and the pollen-collecting ability of different species of bees, the flowering traits and the pollen loaded on bees were measured in this study. We found that the bees visited flowers with preference and they prefer to forage the flowers having pollen with higher viability. The *B. terrestris* carried more pollen (27,000.00 ± 5613.70 grains) on the body surface than that of the *A. mellifera* (7690.00 ± 2873.26 grains). The *A. mellifera* outperformed *B. terrestris* both in pollen deposition per visit on the stigma and on the pollen transfer efficiency. The viability of pollen on the body surface (67.77% ± 19.06%) and hind legs (92.02% ± 10.74%) of *A. mellifera* were both significantly higher than that of *B. terrestris* (31.84% ± 4.84% and 83.77% ± 6.40%). Our study indicated that the quantity and quality of pollen loaded on stigma by *A. mellifera* were both better than *B. terrestris*, which provided evidence that the *A. mellifera* was the effective pollinator pollinating for the oil tree peony in the pollination net room.

**Keywords:** oil tree peony; pollen viability; pollen deposition; bee pollination





## 1. Introduction

Over 75% of crops that are essential to human daily life rely upon animal pollination to varying degrees [1]. As a pollen 'porter', these pollinators play an irreplaceable role in fertilizing and reproduction of flowering plants [2,3], but also provide huge economic value for humans in the global agriculture [4–6]. Furthermore, there is a closely related connection between plants and pollinators [7]. Most of these pollinators in ecosystem services are insects, especially the bees that have been widely applied in agriculture of more than 200 countries for many years, including the production of fruit, vegetable, and seed [1,8]. Many research works have shown that the quality and quantity of fruits, nuts, and oils were greatly improved by bee pollination both in field and greenhouses conditions [6,9]. As the single most popular species of pollinator for crops worldwide, the Western honey bees (*Apis mellifera* L.) provided huge value in the field of honey production and crop pollination, and they were considered vital pollinators because of the wide availability and effectiveness [6,10,11]. However, it is not always the most effective pollinator in some crops or conditions [12,13]. The bumble bees (*Bombus terrestris* L.), another important pollinator

for agriculture worldwide, has been indicated that they possess a strong adaptation to different climates and habitats, particularly the ability to continue foraging flowers even in high and low temperatures [14], which was one of the reasons making them performance better than honey bee when pollinating the same crop. For instance, few types of research conveyed that the bumble bees outperform the honey bees in pollen deposition, ovary development, and the fruit set, etc. [3,12,13]. Particularly, the ability to collect pollen between the honey bee and bumble bee is also different [15], to a certain extent, which affects the number and viability of pollen on the stigmas that are crucial to the fertilization of angiosperm plants.

Tree peony (*Paeonia suffruticosa* Andr.), a perennial deciduous woody shrub native to China, is mainly used as an ornamental and medicinal plant for approximately 2000 years [16–19]. With the deepening of research on tree peonies in China, studies have shown that the oil yield of its seeds reached 20% and the seed oil was rich in unsaturated fatty acids (UFAs > 90%), especially the indispensable α-linolenic acid that human body cannot synthesize itself [20, 21]. Research on the oil characteristics of tree peonies has made continuous progress in recent years. Nowadays, those tree peony cultivars with strong seed-setting ability, which can be used to produce seeds and high oil yield, were called oil tree peonies. The oil tree peony mainly planted in China contains two series: the oil tree peony 'Feng Dan' (*Paeonia ostii* T. Hong et J.X. Zhang) and the blotched oil tree peony (*Paeonia rockii* T. Hong et J.J. Li) [22], of which the *P. ostii* series has a large amount of flowers, fruits, and strong ecological adaptability. It has been widely planted in more than 20 provinces of China [23]. Developing the oil tree peony industry is of great significance to promote grain and oil production, and to ensure grain and oil safety. However, the low overall yield is still an urgent problem needed to be solved. The current existing research mainly focused on screening of excellent varieties [24–26] and site conditions [27–29], but there are few reports on the pollination of oil tree peony.

The tree peony is a cross-pollinated plant with insects as the main medium, which is partially self-compatible and has a very low self-pollination rate, and the pollinators mainly include bees, flies, beetles, etc., among which bees are the main pollinators [30–35]. The number of mature fruits and seeds of angiosperms is much lower than the number of flowers and ovules. The reason mainly includes two aspects: one is pollen limitation, that is, there is not enough pollen to transmit to the stigma; the second is resource limitation, that is, not enough nutrients are available for fruit and seed development and maturation [36]. Under natural conditions, the seed rate of *P. ostii* is about 52.16% [37], which indicates that nearly half of the ovules fail to be fertilized or fertilized but fail to develop into seeds. However, the actual pollination of *P. ostii* is insufficient in production practice due to the destruction of the environment and the lack of pollinator resources, which reduces the seed rate and seriously affects the yield [22]. Therefore, solving the pollen limitation of *P. ostii* during the flowering period is the key to greatly increase the yield of *P. ostii*. Based on this, our group conducted a series of studies on bee pollination of *P. ostii* in the early stage. The results suggested that the seed rate was significantly improved by supplementing managed bees during the flowering period of *P. ostii* in the pollination net room [37]. Compared with the blank control, the yield of *P. ostii* pollinated by *A. mellifera* and *B. terrestris* during the flowering period separately increased by 71.84% and 31.88% [38]. The difference in the quantity and quality of pollen on the stigma was the important factor making the *A. mellifera* outperform *B. terrestris* in increasing the seed rate [38]. Thus, the aims of this present study were to determine the pollen-collecting abilities, pollen transfer efficiency, and pollen viability of the bodies of the two species of bees. We discuss the number of pollen on different parts of bees in relation to foraging behavior and seek to better understand the interaction between bees and oil tree peonies, and provide the basis for further studies that the effect of the fertilization process by bees in the oil tree peony.



## 2. Materials and Methods

### 2.1. Study Site

The test sample was collected from the East Garden (34°38′12″ N, 112°39′64″ E) in Luoyang, Henan province, China, corresponding to a warm temperate continental monsoon climate. From 12 April to 27 April in 2022 flowering stages, the ambient temperature ranged between 10–34 °C, average annual rainfall for this location was 611.2 mm.

### 2.2. Experiment Design

Pollination net rooms were set in parallel at the east-west direction of the central location of the field. Each net room was 40 m in length and 8 m in width with a side wall of 2.1 m height and a roof ridge of 3.2 m height. The nylon polyethylene mesh (1 mm) was used to construct the walls and ceilings of the pollination net rooms.

The *P. ostii* was planted in rows 0.9 m apart and spaced 0.9 m apart within the rows. Normal horticultural care was applied in the pollination net rooms. When about 5% of the flowers bloomed in the pollination net rooms, 3 beehives of *A. mellifera* (8000 workers and a queen bee per hive) and 3 beehives of *B. terrestris* (80 workers and a queen bee per hive) were placed in two pollination net rooms, respectively, and we also provided a sugar feeder (50% sugar content) for each beehive to supplement their nectar intake. The *A. mellifera* bees were provided by Luoyang Baishan Apiary and the *B. terrestris* bees were provided by JAHE YASO Eco-technology Co., Ltd.

### 2.3. Flower Trait Measurement

A total of 10 flower buds were randomly selected and tagged to observe and record the following floral traits: petals, calyxes, bracts, stamens, and pistils. Further, 5 flower buds were randomly selected and tagged to observe the flowering dynamics of a single flower. Starting at 10:00 h on the day before blooming, and the state of the flowers was photographed every day until the flowers withered.

### 2.4. Definition of Flowering Stages

The flowering process was divided into three stages according to the time after flowering and the degree of flowering and stamens cracking: the early flowering stage, the peak flowering stage, and the late flowering stage. The corresponding flowers at each stage were respectively called flowers at the early stage (FES), flowers at the peak stage (FPS), and flowers at the late stage (FLS) [32,39].

The early flowering stage: from 1 day after flowering (DAF) to 2 DAF. The flowers began to bloom gradually from the inside out, with the stigmas and anthers exposed. A little pollen was released from anthers at the stage (Figure 1B,C). The peak flowering stage: from 3 DAF to 4 DAF. Flowers were fully blooming at this stage. Most of the anthers were cracked while lots of pollen was released (Figure 1D,E). The late flowering stage: from 5 DAF to 6 DAF. The petals began to wither and the anthers wilt and shrink (Figure 1F,G).

### 2.5. Pollen Load on Bees

To compare the pollen-collecting ability between the *A. mellifera* and the *B. terrestris*, we measured the total pollen loaded on the body surface and pollen loaded on five parts of the two species of bees: the head, thorax, abdomen, forelegs, and midlegs. 5 specimens of each species of bee (the corbicula contained pollen) were haphazardly collected when they were visiting flowers. In the laboratory, a bee sample was divided into five parts under the stereomicroscope, and each part was separately put into a 1.5 mL centrifuge tube with 1 mL of 20% sodium hexametaphosphate solution [40,41]. Each body part of bee was removed after the pollen on the five body parts shaken by the vortex oscillator for 3 min was mixed with the solution. Then, 800 μL supernatant liquid in each centrifuge tube was taken away with a pipette gun after the pollen mixture was centrifuged. The remaining 200 μL pollen mixture represented a sample of pollen on each body part of the bees. Each body part was repeated 5 times (25 samples in total for every species of bee). 1 μL pollen mixture from a

sample was absorbed on the slide with a pipette after the pollen mixture was shaken by the vortex oscillator and then the cover glass was covered. The number of pollen grains in each slide was counted using the microscope. Each sample was repeated 5 times (5 slides in each sample). And the total pollen loaded on the body surface of the *A. mellifera* and the *B. terrestris* was calculated.

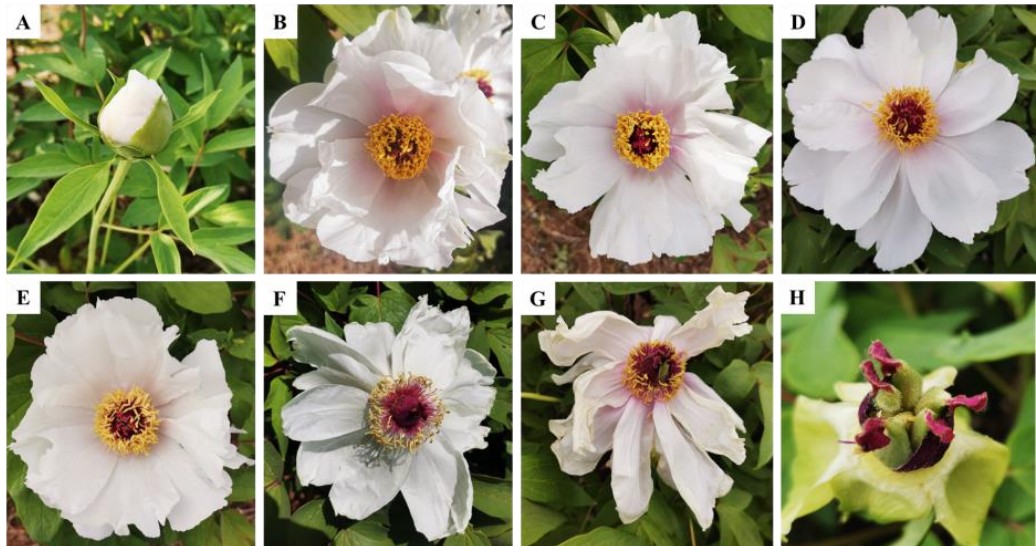

**Figure 1.** The flowers of *P. ostii* at different stages (**A**) Before flowering (0 DAF), bud; (**B**) 1 DAF, flowers at the early stage; (**C**) 2 DAF, flowers at the early stage; (**D**) 3 DAF, flowers at the peak stage; (**E**) 4 DAF, flowers at the peak stage; (**F**) 5 DAF, flowers at the late stage; (**G**) 6 DAF, flowers at the late stage; (**H**) 7 DAF.

### 2.6. The Pollen Quantity of P. ostii per Flower and Pollen Deposition per Visit

A total of 10 unopened flowers were randomly selected and taken back to the laboratory. The number of stamens contained in each flower was counted and recorded in the laboratory. 5 uncracked stamens in each flower were separately put into a 5 mL centrifuge tube. Adding 4 mL of 20% sodium hexametaphosphate solution into each centrifuge tube after they were dried in the baking oven [40]. The stamens were removed after the pollen in the stamens was shaken by the vortex oscillator for 3 min. Then, 3 mL supernatant liquid in each centrifuge tube was taken away with a pipette gun after the pollen mixture was centrifuged, and 1 μL pollen mixture was dropped on the slide with a pipette after the pollen mixture was fully shaken by vortex oscillator and covered the cover glass. The number of pollen grains in each slide (5 slides for each flower) was all counted using the microscope and then the total amount of pollen per flower was calculated.

To compare pollen transfer efficiency between the two species of bees, the pollen deposition per visit was measured. Flowers that had not yet opened were selected from ten plants and bagged with fine nylon mesh nets. The bags were removed when the flowers began to bloom, and the flowers visited by a bee that first contacted the stigma were immediately sampled. The five stigmas in each flower were all put into a 1.5 mL centrifuge tube with 20% sodium hexametaphosphate solution. The subsequent operation method of counting pollen deposition per visit was the same as the method of pollen load on bees above.

### 2.7. The Pollen Viability

Pollen viability was tested by TTC (2,3,5-triphenyl tetrazolium chloride) staining method [42]. A total of 40 flower buds blooming the next day were randomly selected and tagged. Several stamens from 5 tagged flowers were taken into 1.5 mL centrifuge tubes at 10:00 h on the day before blooming, 1st, 2nd, 3rd, 4th, 5th, and 6th day after flowering. In

the laboratory, stamens on each day after flowering were separately taken into a 1.5 mL centrifuge tube with 1 mL TTC staining solution. Taking away stamens after the pollen was fully mixed with a solution by using the vortex oscillator. Part of the pollen mixture was drawn on the slide with a pipette and then cover with the cover glass. Observing the dyeing situation of pollen using the microscope after incubation in a 25 °C incubator for 15 min. Three repeats of the samples from each day and five visual fields were randomly selected from each slide. The red pollen grains were viable, and the faint-red pollen grains were weakly viable while the unstained pollen grains were non-viable (Figure 2). The proportion of viable pollen in the total number of pollens per visual field was calculated as the viability of pollen.

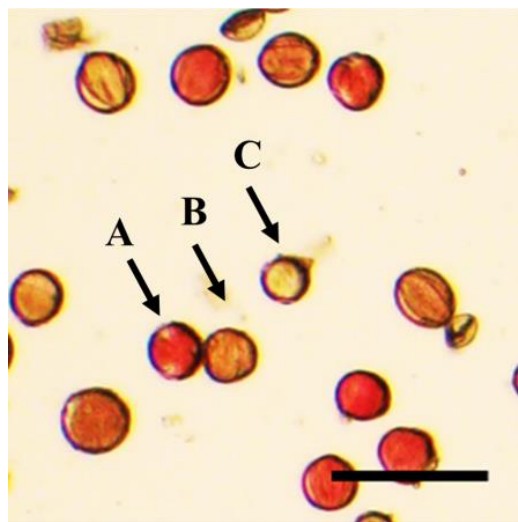

**Figure 2.** The dyeing situation of pollen of *P. ostii* with TTC staining method. (A) pollen with strong viability; (B) pollen with weak viability; (C) pollen without viability; Scale bars, 100 μm.

A total of 5 flowers at the early stage, the peak stage, and the late stage were separately selected and several stamens at each stage were taken into 1.5 mL centrifuge tubes with 1 mL TTC staining solution. Subsequently, the test method of pollen viability at different flowering stages is consistent with the above method.

A total of 5 specimens of each species of bee (the corbicula contained pollen) were haphazardly collected when they were visiting flowers. Each bee body was divided into two main parts (body surface and hind legs) with the stereomicroscope in the laboratory. The two main parts were placed in 1.5 mL centrifuge tubes with 1 mL TTC staining solution. Also, the test method for the viability of pollen on the two main parts of bees consistent with the above method.

*2.8. Data Analysis*

All statistical analyses were performed using IBM SPSS 23.0 (Chicago, IL, USA). A Shariro-Wilk normality test was used to test normality and a Levene test was used to test homoscedasticity before using independent-sample *t*-tests and the one-way analysis of variance followed by a least significant difference LSD multiple comparisons.

**3. Results**

*3.1. Observation of Floral Traits*

One flower on the top of the mature branches of *P. ostii*, with white petals and a pale pink center of some flowers (Figure 3A). Radial corolla with 1–2 rounds petals, 9–13 petals; 2–4 oval or spoon-shaped Calyxes; 4–6 green lanceolate bracts; 145–245 polyadelphous stamens with mauve filament and yellow anther (Figure 3B); 3–8 carpels, 3–18 ovules

per carpels; purple stigma on the top of pistil, its spine secretes colorless and transparent mucus; green ovary, the outer part was covered with a purplish-red tunic (Figure 3C).

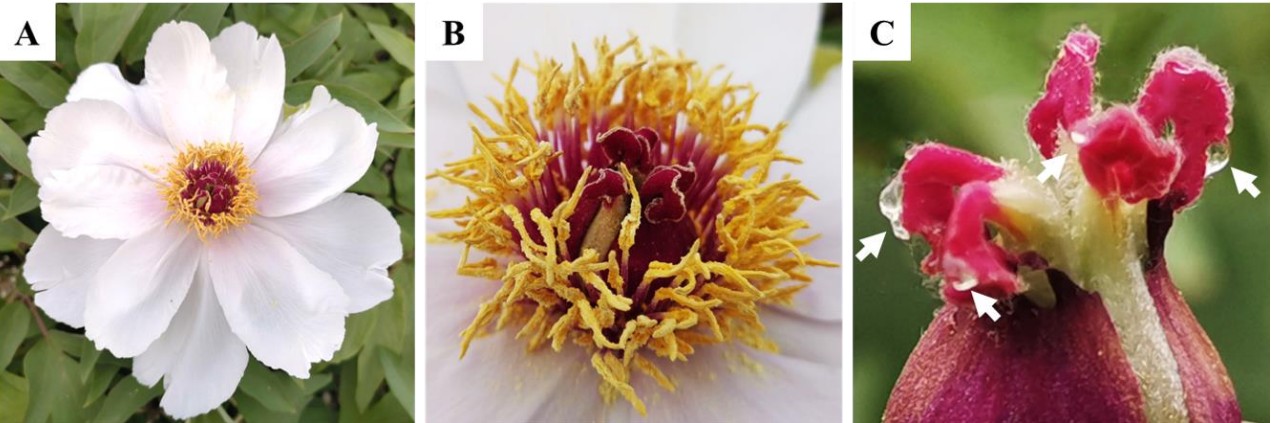

**Figure 3.** Floral traits of *P. ostii*. (**A**) Whole flower; (**B**) Anthers dehiscence; (**C**) Mucus secreted from the stigmas.

### 3.2. Pollen Loads on Bees

With rich fluff, more pollen was adhered to the body of bees when they foraged flowers. The result revealed that the *B. terrestris* carried more pollen grains (26,280.00 ± 10,244.95 grains, mean ± SD) on its body surface than that of the *A. mellifera* (7690.00 ± 2873.26 grains) (Figure 4, t = 3.502, $p < 0.05$).

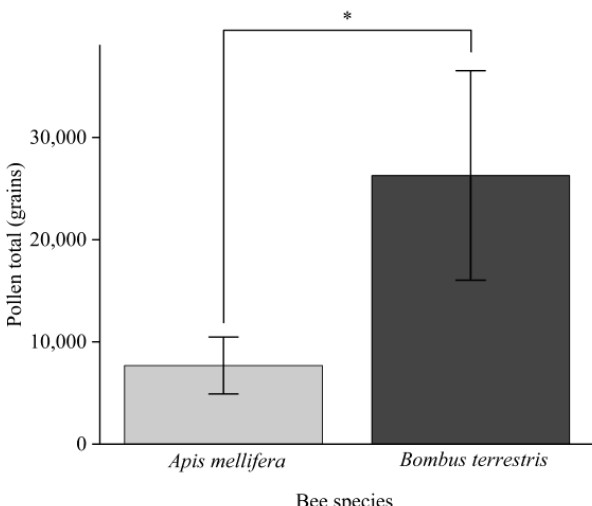

**Figure 4.** The total pollen loading on the body surface of the *A. mellifera* and the *B. terrestris*. * represents that there are significant differences in the total pollen on the body surface between the *A. mellifera* and the *B. terrestris* (Student's *t*-test, $p < 0.05$).

Further, pollen loads on the five body parts of *B. terrestris* were all significantly higher than that of the *A. mellifera* (Figure 5, $p < 0.05$), and the variation of pollen quantity carried by *A. mellifera* and *B. terrestris* in five parts was different. The thorax and midlegs of *A. mellifera* carried more pollen (1816.00 ± 1495.46 grains and 2790.00 ± 1179.61 grains, respectively, mean ± SD) than the head, abdomen and forelegs of *A. mellifera* (1016.00 ± 852.29 grains, 720.00 ± 416.33 grains, and 696.00 ± 513.55 grains, respectively) (Figure 5, F = 4.412, df = 4, $p < 0.05$). The thorax, abdomen, and midlegs of *B. terrestris* carried more pollen grains (7088.00 ± 3601.43 grains, 5950.00 ± 4363.42 grains, and 8248.00 ± 3037.70 grains,

respectively) than the head and forelegs of *B. terrestris* (2704.00 $\pm$ 914.91 grains and 3040.00 $\pm$ 1315.29 grains, respectively) (Figure 5, F = 3.823, df = 4, $p < 0.05$).

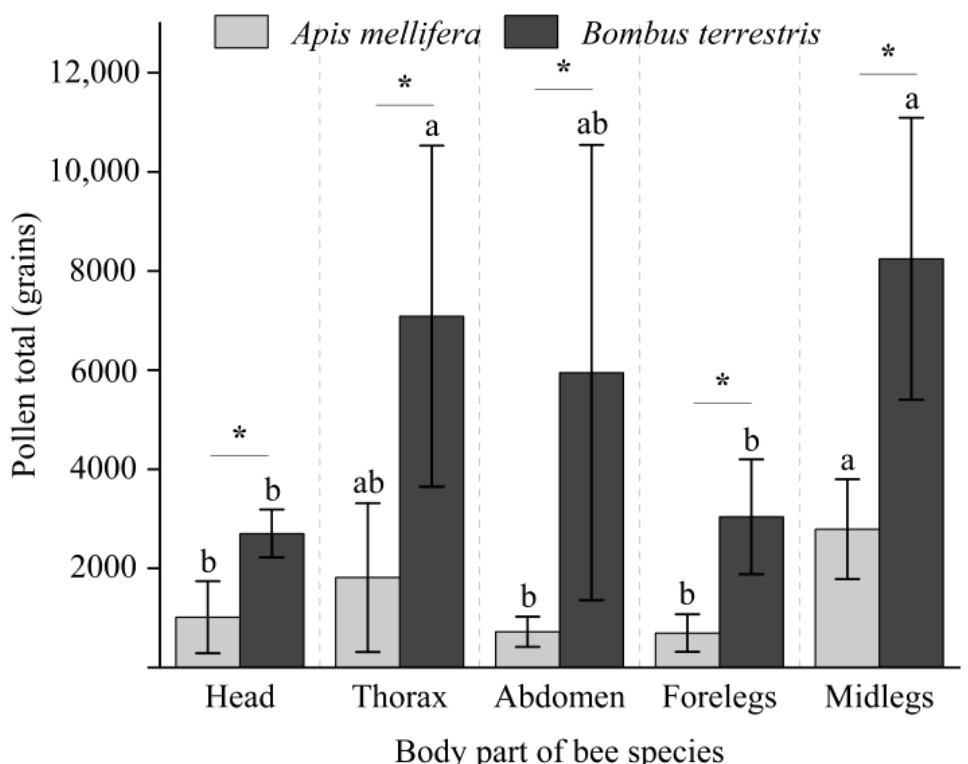

**Figure 5.** The total pollen loaded on the different body parts of the *A. mellifera* and the *B. terrestris*. Different letters indicate significant differences among the body parts of the *A. mellifera* and *B. terrestris* (One-way Anova, $p < 0.05$). * represents that there are significant differences in the number of the same body part between the *A. mellifera* and the *B. terrestris*. (Student's *t*-test, $p < 0.05$).

### 3.3. Pollen Loads on Bees

The pollen quantity per flower of *P.ostii* was $1.61 \times 10^7 \pm 3.66 \times 10^6$ grains (n = 10). The *A. mellifera* and *B. terrestris* differ significantly in terms of pollen deposition on stigma per visit. Pollen deposition per visit mediated by *A. mellifera* (27,000.00 $\pm$ 5613.70 grains, mean $\pm$ SD) was significantly higher than that by *B. terrestris* (7746.67 $\pm$ 3089.68 grains) (Table 1, t = 5.204, $p < 0.05$). The pollen transfer efficiency of *A. mellifera* (0.17%) was higher than that of *B. terrestris* (0.05%) (Table 1).

**Table 1.** Comparisons of pollen deposition per visit and pollen transfer efficiency (mean $\pm$ SD, n = number of sampled flowers) between *A. mellifera* and *B. terrestris* in *P. ostii*. Different letters indicate significant differences between the two species of bees (Student's *t*-test, $p < 0.05$).

| Bee Species | Pollen Deposition per Visit | Pollen Transfer Efficiency (%) |
|---|---|---|
| *A. mellifera* | 27,000.00 [a] $\pm$ 5613.70 (n = 3) | 0.17 |
| *B. terrestris* | 7746.67 [b] $\pm$ 3089.68 (n = 3) | 0.05 |

### 3.4. Pollen Viability of Different Days after Flowering

From 1 DAF to 6 DAF, the stamens gradually unfold from the inside to the outside. The anther gradually cracked from top to bottom while the pollen was released. On 4 DAF, the stamens began to wilt and fell off, and the stamens basically all fell off on 7 DAF (Figure 6A).

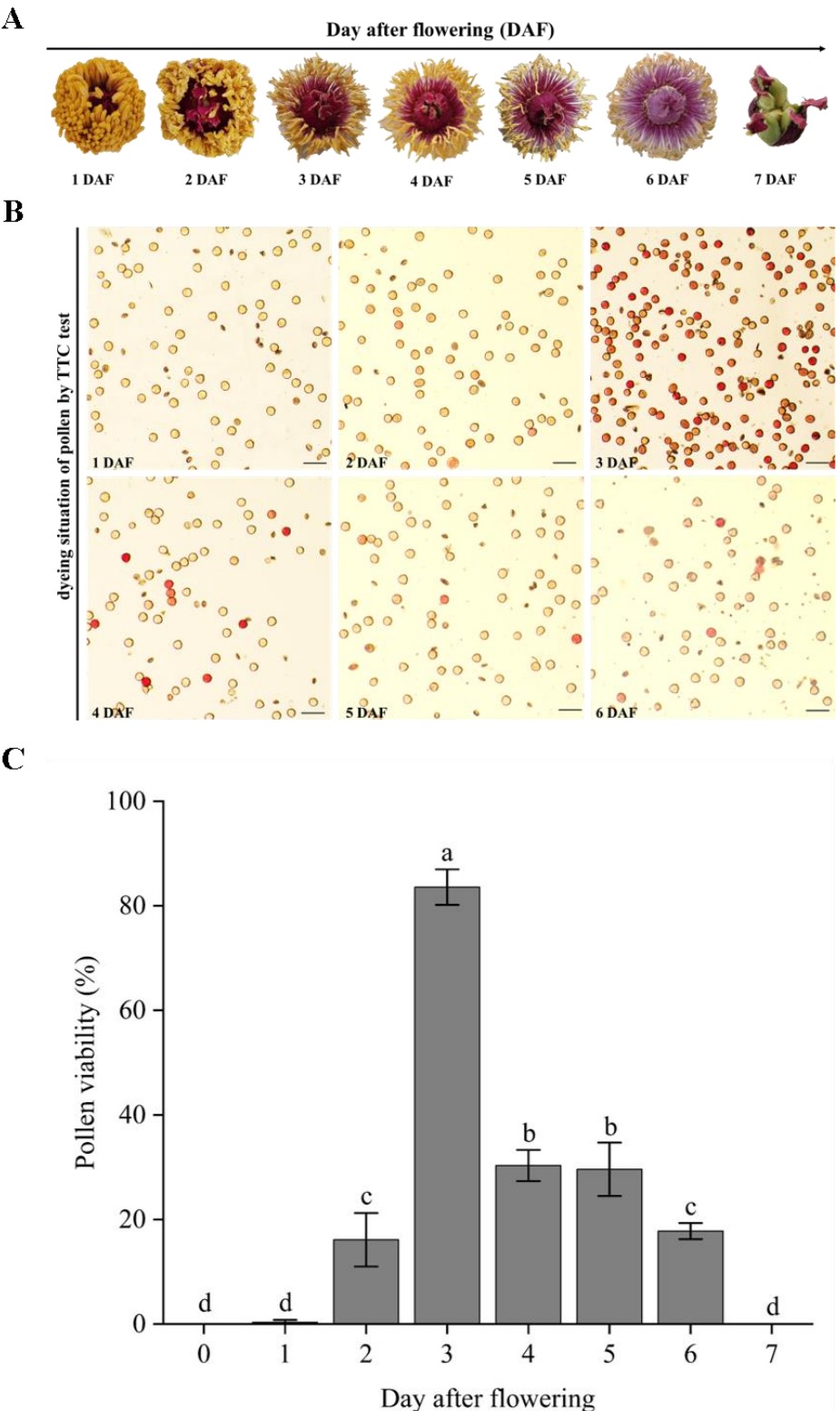

**Figure 6.** Dynamic changes of androecium and stamens of *P. ostii* during 7 consecutive days after flowering. (**A**) Dynamic changes of androecium during 7 consecutive days after flowering. (**B**) The dyeing situation of pollen by TTC test during 6 consecutive days after flowering, scale bars, 100 μm. (**C**) Dynamic changes of pollen viability of *P. ostii* during 7 consecutive days after flowering. Different letters indicate a significant difference in the pollen viability on different days after flowering (One-way ANOVA, $p < 0.05$).

It's demonstrated that the pollen viability of *P. ostii* was significantly different at different flowering days and a trend of increasing first and then decreasing from 0 DAF to 7 DAF (Figure 6B,C). Before flowering (0 DAF), the anther has not yet cracked and the pollen shaken by the vortex oscillator has shown no viability. On 7 DAF, stamens have fallen off and their pollen viability was counted as zero. The pollen viability on 3 DAF was the highest (83.58% ± 3.39%, mean ± SD), which was significantly higher than that on 4 DAF (30.33% ± 2.98%) and 5 DAF (29.59% ± 5.10%) (F = 246.433, df = 7, *p* < 0.05) (Figure 6C). Pollen viability began to decrease on 4 DAF, which was consistent with the changing trend of stamen wilting and falling off. The pollen was active for 6 days (Figure 6C).

### 3.5. Pollen Viability at Different Flowering Stages

There were significant differences in the pollen viability at different flowering stages, of which the pollen viability of the flowers at the peak stage (56.95% ± 0.63%, mean ± SD) was the highest, followed by the flowers at the late stage (23.68% ± 2.26%), and the flowers at the early stage was the lowest (8.23% ± 2.34%) (Figure 7). Further, the pollen viability of flowers at the peak stage was significantly higher than that of flowers at the early stage and flowers at the late stage (Figure 7, F = 508.991, df = 2, *p* < 0.05).

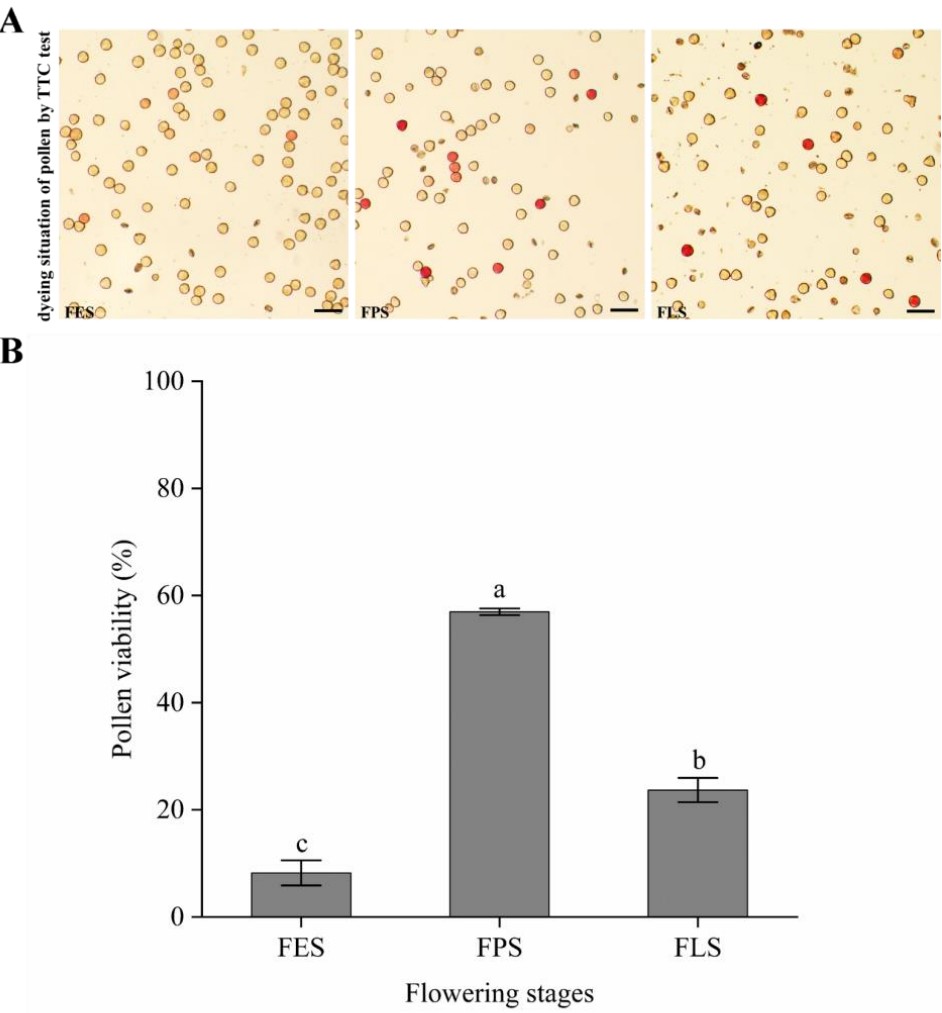

**Figure 7.** Pollen viability of *P. ostii* at different flowering stages. (**A**) The dyeing situation of pollen at different flowering stages by TTC test, scale bars, 100 μm. (**B**) Pollen viability of *P. ostii* at different flowering stages, FES: flowers at the early stage; FPS: flowers at the peak stage; FLS: flowers at the late stage. Different letters indicate a significant difference in the pollen viability among different flowering stages (One-way ANOVA, *p* < 0.05).

### 3.6. Pollen Viability Carried by Body Surface and Hind Legs of Different Pollinators

For different species of bees, the viability of pollen on the body surface (67.77% ± 19.06%, mean ± SD) and hind legs (92.02% ± 10.74%) of *A. mellifera* were both significantly higher than that of *B. terrestris* (31.84% ± 4.84% and 83.77% ± 6.40%, respectively) (Figure 8A,B, t = 4.084, $p < 0.05$ and t = 1.472, $p < 0.05$). In addition, the viability of pollen carried by the hind legs of *A. mellifera* and *B. terrestris* was significantly higher than that of the body surface (Figure 8A,B, t = 2.476, $p < 0.05$ and t = 14.473, $p < 0.05$).

**A**

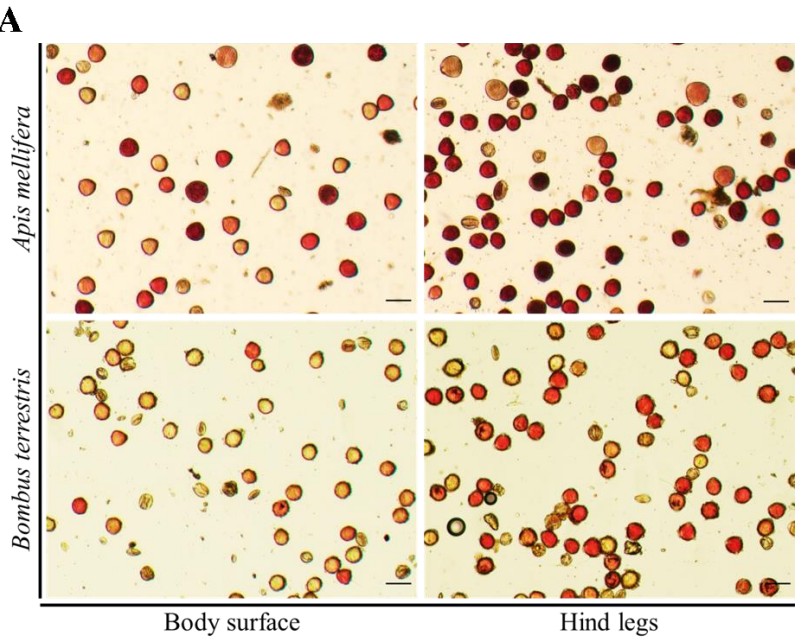

**B**

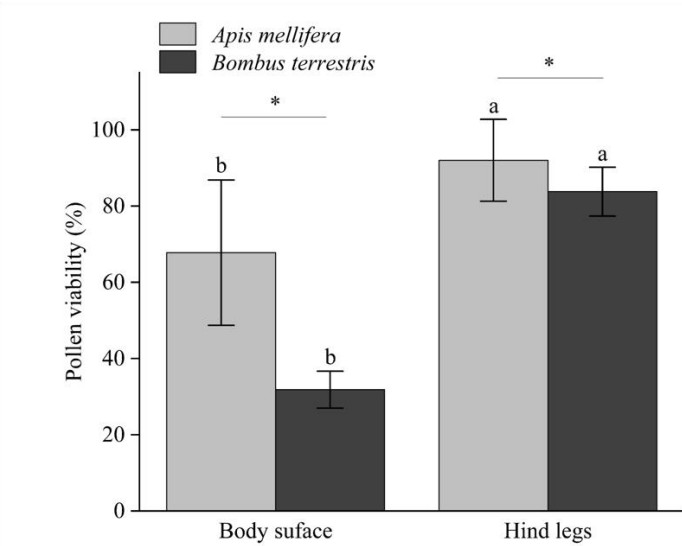

**Figure 8.** The viability of pollen on the body surface and hind legs of the two species of bees. (**A**) The dyeing situation of pollen carried by the body surface and hind legs of the two species of bees. Scale bars, 50 μm. (**B**) The viability of pollen on the body surface and hind legs of the two species of bees. Different letters indicate significant differences between the body surface and hind legs within the *A. mellifera* and the *B. terrestris* (Student's *t*-test, $p < 0.05$). * represents a significant difference between the *A. mellifera* and the *B. terrestris* in the viability of pollen on the body surface and hind legs (Student's *t*-test, $p < 0.05$).

## 4. Discussion

The number of pollens on bees was related to the visit duration of per flower [41,43]. Compared with pollinators taking more time at a single flower, those pollinators taking less time at a single flower usually visit more flowers in the same period of time, which makes them carry more pollen on their body [41]. Our results in the study also illustrated that pollen loads on the body surface of the *B. terrestris* were significantly higher than that of the *A. mellifera*, consistent with our early study that the *A. mellifera* of the single visitation duration and the total visitation duration per flower were both longer than that of the *B. terrestris* [37,44]. Besides, a key factor needed to be taken seriously is the difference in the body size between the two species of bees. It is obvious that the body size of *B. terrestris* was larger than *A. mellifera*. In total, in terms of the same species of pollinators, the larger the size, the greater the amount of pollen carried by pollinators [45], but sometimes it was the opposite [46]. The number and shape of the hair of pollinators were another two important factors influencing the number of pollens on the body surface [47]. Whether the *A. mellifera* or the *B. terrestris*, they were both hairy and had the same species of hairstyle. Therefore, the number of pollens on the body surface of different species of bees was affected by multiple factors, and further research is needed as to determine which factor plays the key role. In the study of potential pollinators in an understory dioecious shrub (*Helwingia japonica* Thunb. F. Dietr), the pollens were distributed evenly on a different part of their bodies [48]. However, the pollen placement on the different parts of bees was significantly different in our study, which makes the phenomenon may be related to the behavior of grooming pollen. In the field, we observed that the bees constantly wipe the body with their legs during collecting pollen, and our results also conveyed that the pollen on the midlegs was the most.

In our preliminary study, the result showed that the seed rate and the growth rate of *P. ostii* were significantly improved by supplementing the *A. mellifera* and the *B. terrestris* during the flowering period in the pollination net room, which conveyed that they were both considered to be the effective pollinators of oil tree peony [37,38,44,49,50]. We measured pollen deposition per visit and compared the pollen transfer efficiency of the two species of bees in this study, and the result showed that the *A. mellifera* was the most effective pollinator. The *A. mellifera* provided enough pollen to guarantee every ovule in the ovary to be fertilized owing to more transferred pollen to the stigmas. In contrast, in the study of peach, the *B. patagiatus* deposited more pollen on stigmas than the *A. mellifera* both in a single visit and one day of open pollination [3]. Several factors may explain this difference. First, the pollen-collecting ability was different in the different species of bumble bees due to the difference of foraging behavior and the size [51]. Second, the number of pollens on the body surface was also affected by the flora characteristics of plants [52], such as the size of the whole flower, the number of stamens, pistils and pollen grains, and the position relation between stamen and pistil, etc., what was called the interaction relationship between plants and pollinators [53,54].

Pollen viability refers to the ability to survive and germinate [55,56]. Our results showed that the pollen viability of *P. ostii* increased first and then decreased during the flowering period. On the 3rd day after flowering, the pollen viability was the highest, and the pollen had viability was up to 6 days, which was basically consistent with the trend of pollen viability of *Paeonia decomposita* [32] and *Paeonia ludlowii* [35], but it was different from the results of Luo et al. [33] on the pollen viability of *P. ostii*. The genetic characteristics of pollen and external factors are important factors affecting its viability [57].

The pollen viability of *P. ostii* was different at different flowering stages. Our result conveyed that the pollen viability was the highest at the peak flowering stage, followed by the late flowering stage, and the lowest at the early stage. The anthers were basically all mature and cracked at the peak flowering stage, and a large amount of pollen having viability was released. We found that the viability of pollen carried by the body surface of *A. mellifera* was significantly higher than that of flowers at the peak stage under natural conditions, but the viability of pollen carried by the body surface of *B. terrestris* was lower

than that of flowers at the peak stage. Also, the viability of pollen loaded on the body surface of *A. mellifera* was higher than that of the *B. terrestris*, which might be due to the higher evolution degree of the *A. mellifera* [39]. In the field, we observed that the *A. mellifera* generally fell on the androecium of flowers, and then judge whether the pollens were suitable for collection, and the *A. mellifera* preferred to visit flowers possessing highly viable pollens while the *B. terrestris* usually concentrated on collecting pollen only if they fell on the androecium no matter how the pollen viability was high or low. Generally, the pollens in an anther were released after they were mature, resulting that the viable pollens were adhered to the body surface of the bees when they forage flowers. Furthermore, the whole anthers including immature and abortive pollens were sampled when measuring the viability of pollen of *P. ostii*.

The pollen the bees collected was stored on the corbicula and mixed with nectar or the salivary gland secretion of the bees [58,59]. Except that the pollen is as a source of nutrients for adult bees and larvae, it also provides a healthy function for human beings and shows huge potential economic value because it contains more than 200 nutritional ingredients, including various amino acids, vitamin and microelement, etc. [58,60]. The bees in the pollination net room only collected pollen due to the *P. ostii* without nectar [33,44]. Our study indicated that the pollen on the hind legs of both the *A. mellifera* and the *B. terrestris* was significantly higher than that on the body surface, which illustrates that the viability of pollen in the corbicula was possibly influenced by the saliva of bees or the fusion constituent between the pollen and the saliva. In addition, the viability of pollen on the hind legs of the *A. mellifera* was significantly higher than that of the *B. terrestris*.

## 5. Conclusions

In conclusion, the *B. terrestris* outperformed the *A. mellifera* in the ability of pollen-carrying. Conversely, the pollen loaded on *A. mellifera* was significantly higher than *B. terrestris*, and the pollen transfer efficiency of *A. mellifera* was significantly higher than that of *B. terrestris*. In addition, to understand why the seed rate was improved by bees, our study provides evidence that the *A. mellifera* was outperform *B. terrestris* in pollinating pollen. Thus, it's vital for supplying bees in the oil tree peony field if there are few wild pollinators, and the *A. mellifera* was the effective pollinator under the pollination net rooms.

**Author Contributions:** Conceptualization, J.B., K.Z. and X.H. (Xiangnan He); methodology, C.H. and X.H. (Xiaogai Hou); validation, C.H. and X.H. (Xiaogai Hou); formal analysis, J.B.; investigation, J.B., Z.C. and J.W.; data curation, J.B.; writing—original draft preparation, J.B.; writing—review and editing, J.B. and K.Z.; visualization, X.H. (Xiaogai Hou); project administration, X.H. (Xiaogai Hou); funding acquisition, X.H. All authors have read and agreed to the published version of the manuscript.

**Funding:** This research was funded by the Innovation Scientists and Technicians Troop Construction Projects of Henan Province, grant number 212101510003 and the Luoyang Rural Revitalization Project, grant number 2101099A.

**Data Availability Statement:** The data presented in this study are available on request from the corresponding author. The data are not publicly available due to restrictions privacy.

**Acknowledgments:** Authors are very grateful to Xian Xue and Qi Guo for providing suggestions on the experimental design, Ming Huang for providing laboratory instruments, Wanqing Zhang and Yasang Luo for helping the lab work.

**Conflicts of Interest:** The authors declare no conflict of interest.

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
