# Peer review of "Comparison of Pollen-Collecting Abilities between Apis mellifera L. and Bombus terrestris L. in the Oil Tree Peony Field"

_horticulturae, doi:10.3390/horticulturae9060658_

Round 1

Reviewer 1 Report

 This paper compares the pollination efficiency of Apis mellifera and Bombus terrestris for oil peony flowers. The results showed that A. mellifera had higher pollen-carrying capacity and pollination efficiency than B. terrestris. This is an interesting paper, and the results are very good. It has important implications for the application of pollination in this crop.

Although it can be expected that the pollination efficiency of A. mellifera is better than that of B. terrestris, but the use of a honeybee colony contains thousands of individuals, and the bumblebee colony only hundreds of individuals. The author did not mention the colony status of the two species, it is very curious whether there is a relevant difference for pollination in the number of two bees in the net-room.

The author has worked hard in the article, and the quality of the article is commendable. It is recommended that the author make appropriate corrections before accepting this manuscript, the suggested result is as follows:

L20,21,23,24 (Apis mellifera L. and Bombus terrestris L.) à Repeated use. “L.” can be omitted

L61 62 Paeonia ostii T. Hong et J.X.Zhang, Paeonia rockii T. Hong & J.J.Li à For multiple authors, use "et" or "&" consistently

Fig 4 Y axis à add unit; x axis à use term “bee species” rather than “different types of bees”

Fig 5 Y axis à add unit; x axis à use term “body parts of bee species” rather than “different body parts of bees”

Figure 7B. use term “Flowering stages” rather than “different flowering stages”.

L 322 Helwingia japonica,à add author(s)

L 385-387 “There were significant differences in the amount of pollen on different parts of the two types of bees.” à Ambiguity sentence

L109-114 “3 beehives of A. mellifera and 3 beehive” à Please explain the internal conditions of the bee colony used, such as the number, presence or absence of queen bees, feeding or management methods...

Table 1. “Different kinds of bees” à use term “Bee species” rather than “Different body kinds of bees”

Ref 6. Check format!

Ref 13. Check format! Scientific name in italics!

Author Response

Dear reviewer,

Reviewer 2 Report

Dear Editor,

This study aimed to determine the pollen-collecting ability, pollen transfer efficiency, and pollen viability on the body of the two types of bees. This topic can be very interesting for journal readers. The methodology of the manuscript is well done. The manuscript can be accepted for publication after minor suggestions.

 Best regards,

Reviewer

 English language must be significantly improved in the manuscript.

Author Response

Dear reviewer,

Reviewer 3 Report

Manuscript ID: Horticulturae-2399726

The paper entitled “The Apis mellifera L. outperform the Bombus terrestris L.: the quantity and quality of pollen pollinated for the oil tree peony in the pollination net room” was carefully reviewed. The objective of this study is to determine the pollen-collecting ability, pollen transfer efficiency and the pollen viability on the body of A. mellifera and B. terrestris.

This manuscript is an interesting field study but the limitations of the work, novelty, and contributions should be highlighted more.

Detailed comments:

Title

-          The title is too long and technical, when it should be concise and informative.

Abstract

-          Add quantitative results to the Abstract.

-          Lines 20-24: replace “Apis mellifera L. …, Bombus terrestris L.” by “A. mellifera …, B. terrestris…”

Keywords

-          In general, avoid using keywords that are already in the title. Replace “Apis mellifera L.; Bombus terrestris L.; oil tree peony; pollination, …”.

Introduction

-          Line 37: Replace “…bee pollination both in open pollination systems (fields) and enclosed pollination systems (greenhouses)” by “…bee pollination both in field and greenhouse conditions”.

-          Replace “several researches” by “few researches” as you only refer to 3 papers [3,12,13].

-          Line 78: Give a reference for this statement: “the actual pollination of P. ostii is insufficient in production practice due to the destruction of the environment and the lack of pollinator resources, which reduces the seed rate and seriously affects the yield”.

-          Line 87: Give a reference for this statement: “Difference in the quantity and quality of pollen on the stigma was the important factor making the A. mellifera outperform B. terrestris in increasing the seed rate”.

-          Lines 88-94: The last paragraph is too long and complex. It could be split into two sentences: one stating the aims of the study and one stating the implications of the study. This paragraph contains some grammatical errors and unclear expressions: "the pollen-collecting ability" should be "the pollen-collecting abilities", "the pollen viability on the body" should be "the pollen viability on their bodies", "the number of different parts of body the in relation to foraging behavior" should be "the number and role of different body parts in relation to foraging behavior", and "provide the basic for further study" should be "provide the basis for further studies".

Materials and methods

-          Line 112: Replace “with a sugar feeder (50 % sugar content)” by “We also provided a sugar feeder (50 % sugar content) for each beehive to supplement their nectar intake”.

-          Lines 205-209, “Data analysis”: This subsection could be improved by adding some details and clarifications: what were the dependent & independent variables in the T tests and the ANOVA? What was the significance level for the tests? How did the normality and homoscedasticity tests affect the choice of tests?

Results:

-          Lines 211-220, “3.1. Observation of floral traits”: This part is descriptive and informative, but it may not be necessary if the floral traits of P. ostii have already been reported in previous studies. I recommend deleting this paragraph and cite the relevant sources instead in the Materials & methods Section.

-          Line 236, Figure 4: Delete the abscissa title as it is irrelevant: “Different types of bees”.

-          Line 254, Table1: Replace “Different kinds of bees” by “Species”.

-          Line 260: Correct this sentence: “It’s demonstrated showed that the pollen viability of P. ostii was significantly …”

Discussion:

-          Line 321: Rewrite this sentence because it is difficult to understand: “In the study of potential pollinators in Helwingia japonica, these floral insects of H. japonia were considered as opportunistic visitors because pollen were distributed evenly on different part of the visitor’s body”.

-          Line 322: add the common name of Helwingia japonica.

-          Line 381-382: Delete this statement as you have not demonstrated this in the current study: “…which conveyed that difference existed at the composition of saliva and synthetic products between the A. mellifera and the B. terrestris”.

Conclusion

-          The conclusion section is not appropriate because the authors are repeating the same information that they have already presented in the results section. Therefore, I recommend that the authors restate the main results. It would be helpful to add a sentence that explains why the results are important for crop management or pollination ecology.

The manuscript would really benefit from proofreading by an English editing service or a native English speaker as some sentences are very difficult to understand and there are many errors and typos.

The manuscript would really benefit from proofreading by an English editing service or a native English speaker as some sentences are very difficult to understand and there are many errors and typos.

Author Response

Dear reviewer,

Reviewer 4 Report

Dear Authors,

thank you for the opportunity to meet the manuscript entitled: "The Apis mellifera L. outperform the Bombus terrestris L.: the quantity and quality of pollen pollinated for the oil tree peony in the pollination net room".

The subject of this study is very interesting and undoubtedly important in relation to the increase of the pollinator population.

The composition of the manuscript is at a good level, the proportionality of the individual chapters is ensured.

I appreciate the thorough description of the materials and methods used, as well as the performed statistical analyses.

The results are processed and evaluated correctly, and the authors sufficiently confronted them with the knowledge available so far. This is evidenced by the number of cited references.

Author Response

尊敬的审稿人,

请参阅附件。
